

**Long-term projections of global water use for electricity generation under the**
**Shared Socioeconomic Pathways and climate mitigation scenarios**
Nozomi Ando[1], Sayaka Yoshikawa[1], Shinichiro Fujimori[2], and Shinjiro Kanae[1]
[1]Department of Civil and Environmental Engineering, Tokyo Institute of Technology, 2–12–1 O-okayama, Meguro-
ku, Tokyo, Japan
[2]Center for Social and Environmental Systems Research, National Institute for Environmental Studies, 16–2
Onogawa, Tsukuba, Ibaraki 305–8506, Japan.
*Correspondence to:* S. Yoshikawa (yoshikawa.s.ad@m.titech.ac.jp) and N. Ando (ando.n.titech@gmail.com)
**Abstract.** Electricity generation may become a key factor that accelerates water scarcity. In this study, we estimated
the future global water use for electricity generation from 2005 to 2100 in 17 global sub-regions. Twenty-two future
global change scenarios were examined, consisting of feasible combinations of five socioeconomic scenarios of the
Shared Socioeconomic Pathways (SSPs) and six climate mitigation scenarios based on four forcing levels of
representative concentration pathways (RCPs) and two additional forcing levels, to assess the impacts of
socioeconomic and climate mitigation changes on water withdrawal and consumption for electricity generation.
Climate policies such as targets of greenhouse gas (GHG) emissions are determined by climate mitigation scenarios.
Both water withdrawal and consumption were calculated by multiplying the electricity generation of each energy
source (e.g., coal, nuclear, biomass, and solar power) and the energy source-specific water use intensity. The future
electricity generation dataset was derived from the Asia-Pacific Integrated/Computable General Equilibrium
(AIM/CGE) model. Estimated water withdrawal and consumption varied significantly among the SSPs. In contrast,
water withdrawal and consumption differed little among the climate mitigation scenarios even though GHG
emissions depend on them. There are two explanations for these outcomes. First, electricity generation for energy
sources requiring considerable amounts of water varied widely among the SSPs, while it did not differ substantially
among the climate mitigation scenarios. Second, the introduction of more carbon capture and storage strategies
increased water withdrawal and consumption under stronger mitigation scenarios, while the introduction of more
renewable energy decreased water withdrawal and consumption. Therefore, the socioeconomic changes represented
by the SSPs had a larger impact on water withdrawal and consumption for electricity generation, compared with the
climate mitigation changes represented by the climate mitigation scenarios. The same trends were observed on a
regional scale, even though the composition of energy sources differed completely from that on a global scale.

**1. Introduction**

With economic and population growth, energy demand is likely to continue increasing in the coming decades,
and the energy sector is becoming a large water consumer. For example, the global water withdrawal for electricity
generation in 2010 amounted to about 540 km$^3$ yr$^{-1}$, or 14% of the global total water withdrawal (IEA, 2012), while



electricity generation accounted for about 70% of industrial water withdrawal in 2010. There is concern that water
use for electricity generation could increase competition with other major water users, including agriculture,
manufacturing, and domestic users. Furthermore, water shortages could impair energy security. Electricity shortages
have recently been caused by water shortages in the southeastern United States, the Pacific Northwest, and continental
Europe (Bartos et al., 2015). Therefore, it is important to estimate how much water will be required for electricity
generation in the future.
Water use for electricity generation falls under industrial water use. Global industrial water use projections have
been presented by Alcamo et al. (2007), Shen et al. (2008), and Hanasaki et al. (2013). However, these studies did
not differentiate water use for electricity generation from water use for other industrial processes. Vassolo and Döll
(2005) and Flörke et al. (2013) estimated global industrial water use by distinguishing water use for electricity
generation and manufacturing water. However, they used global hydrological models on a grid scale, which are not
designed to readily assess the global impact of demand drivers, such as energy source composition, cooling system
shares, and technological improvements, in the distant future.
There is another approach. Socioeconomic changes and climate mitigation are among the most significant
demand drivers in future projections; the global impact of these demand drivers and others on water use for electricity
generation can be assessed using a global economic model on a regional scale. Most studies using this approach have
focused on only one of these drivers (Kyle et al., 2013; Hejazi et al. 2014; Bijl et al. 2016; Fricko et al. 2016); to our
knowledge, only Fujimori et al. (2016a) has examined both socioeconomic changes and climate mitigation changes.
Fujimori et al. (2016a) estimated the future industrial water withdrawal under the Shared Socioeconomic Pathways
(SSPs; see Sect. 2.1.1) and climate mitigation scenarios based on representative concentration pathways (RCPs; see
Sect. 2.1.1); however, they neither incorporated energy-related factors (e.g., cooling system shares or seawater use
by power plants) nor distinguished water use and withdrawal, which have been taken into account in other studies
using a global economic model on a regional scale.
In this study, we had two objectives: 1) to estimate water withdrawal and consumption for electricity generation
under the SSPs and climate mitigation scenarios based on RCPs for the period from 2005 to 2100 in 17 global sub-
regions while considering energy-related factors, and 2) to compare the impact of the socioeconomic changes and
climate mitigation changes on water withdrawal and consumption for electricity generation, in addition to assessing
each impact. We achieved these objectives by taking advantage of the SSPs and climate mitigation scenarios, which
allowed us to assess the effects of socioeconomic changes and climate mitigation changes separately. In addition to
the SSPs and climate mitigation scenarios, we included assumptions on shifts in the proportion of cooling system
types to assess their potential impacts.
In this study, key drivers of water withdrawal and consumption for electricity generation and scenario settings
are discussed in Sect. 2. The results from the scenario analysis are presented in Sect. 3 and discussed in Sect. 4.
Conclusions are presented in Sect. 5.

**2. Methodology and data**

Water withdrawal and consumption for electricity generation were calculated by multiplying the electricity





generation (MWh) and water use intensity ($m^3$ $MWh^{-1}$) of each energy source. The water use intensity was defined
as water use ($m^3$) per unit electricity generated (MWh). These factors are discussed in Sect. 2.1 and 2.2. We followed
the definitions of water-related terms set by the United States Geological Survey. Water use is defined as the water
used for a specific purpose and includes elements such as water withdrawal and consumption. Water withdrawal is
defined as the water extracted from surface water or groundwater. Water consumption is defined as the proportion of
water withdrawn that is evaporated, transpired, incorporated into products or crops, or consumed by humans or
livestock.

We used future electricity generation data estimated by the Asia-Pacific Integrated Model/Computable General

Equilibrium (AIM/CGE) model, an integrated assessment model developed by National Institute for Environmental
Studies, Japan (Fujimori et al. 2016b). The AIM/CGE model can quantify entire economic goods and service, and
production factors' market exchange with a special focus on energy, agriculture, emissions (GHG and air pollutants)
and land use sectors based on socioeconomic assumptions and climate mitigation targets (e.g., population, gross
domestic product (GDP), and radiative forcing). The impacts of socioeconomic and climate mitigation changes were
assessed using the output of the AIM/CGE model for a target period of 2005–2100; this model covered all regions of
the world, divided into 17 sub-regions (See Table S1and Fig. S1 in Supplementary Information).


**2.1.    Electricity generation**

We used future electricity generation data estimated by the Asia-Pacific Integrated Model/Computable General

Equilibrium (AIM/CGE) model, an integrated assessment model developed by National Institute for Environmental
Studies, Japan (Fujimori et al. 2016b). The AIM/CGE model can quantify entire economic goods and service, and
production factors' market exchange with a special focus on energy, agriculture, emissions (GHG and air pollutants)
and land use sectors based on socioeconomic assumptions and climate mitigation targets (e.g., population, gross
domestic product (GDP), and radiative forcing). The impacts of socioeconomic and climate mitigation changes were
assessed using the output of the AIM/CGE model for a target period of 2005–2100; this model covered all regions of
the world, divided into 17 sub-regions (See Table S1and Fig. S1 in Supplementary Information).

**2.1.1.    Scenario framework**

Future electricity generation was calculated under two sets of scenarios: socioeconomic scenarios and climate

mitigation scenarios. We adopted SSPs to represent the socioeconomic scenarios. The SSPs describe five plausible
future worlds that are defined by narrative storylines and quantitative information and can be characterized by two
indices, socioeconomic challenges for adaptation and for mitigation. In SSP1 (sustainability), both adaptation and
mitigation challenges are low. In contrast, both adaptation and mitigation challenges are high in SSP3 (regional
rivalry). In SSP4 (inequality), adaptation challenge is high but mitigation challenge is low. In SSP5 (fossil-fuel
development), adaptation challenge is low but mitigation challenge is high. SSP2 (middle of the road) falls in an
intermediate position among other four scenarios. The SSPs are described in detail by O'Neill et al. (2014).





The climate mitigation scenarios were represented by six climate mitigation targets and a baseline case. The baseline
case has no constraints on GHG emissions. Meanwhile, the climate mitigation targets consist of four forcing levels
(2.6, 4.5, 6.0, and 8.5 W/m$^2$) of RCPs (van Vuuren et al., 2011), as well as two additional forcing levels (3.4 and 7.0
W/m$^2$). The forcing levels are defined by the cumulative amount of radiative forcing (W/m$^2$) around the year 2100.
Each climate mitigation target is expressed as, for example, the 6.0W case and 2.6W case.

The scenario framework of the socioeconomic scenarios and climate mitigation scenarios is described by

Fujimori et al. (2016b). The baseline cases of SSP1–5 are assumed to correspond to 6.0, 7.0, 7.0, 6.0, and 8.5 W/m$^2$,
respectively. Each combination of SSP and climate mitigation scenario is expressed as, for example, SSP2-6.0W and
SSP3-3.4W.

Figure 1 shows the global electricity generation under the baseline cases for SSP1–5. In all scenarios, global

total electricity generation increased between 2005 and 2100. In particular, the total electricity generation of 2100
was about 7.5 times larger than that of 2005 in SSP5-8.5W. Among the baseline scenarios, renewable energies were
introduced as major energy sources for climate change mitigation in SSP1-6.0W and SSP4-6.0W. Conversely, fossil
fuels were the dominant energy source in SSP3-7.0W and SSP5-8.5W. Meanwhile, nuclear energy grew substantially
over a target period in SSP2-7.0W, SSP4-6.0W, and SSP5-8.5W.

Figure 2 compares the global electricity generation for SSPs and climate mitigation scenarios. Only the baseline

case had a significantly different composition of energy sources from other climate mitigation scenarios in SSP2,
SSP3, and SSP5. In stronger mitigation scenarios, total electricity generation was greater, and more renewable energy
and carbon capture and storage (CCS) were used. The total electricity generation and composition of energy sources
differed greatly between the SSPs within the same climate mitigation scenario. The energy trends and scenario
assumptions are described in detail by Fujimori et al. (2016b).

**2.1.2.    Estimation of electricity generation using freshwater for cooling**

Both freshwater and seawater can be used for electricity generation. However, we focused on freshwater use,

and excluded seawater. We calculated the electricity generation ratio of freshwater- and seawater-based power plants
in the AIM/CGE regions, from which we estimated freshwater-based electricity generation. To calculate the
electricity generation ratio, the electricity generation and water source (freshwater or seawater) for each power plant
around the world are needed. However, we did not have such data. Therefore, we substituted the electricity generation
capacity data of each power plant worldwide for the electricity generation data of each power plant, and assumed that
the electricity generation ratio and electricity generation capacity ratio were nearly the same. The water source of
each power plant was determined based on its distance from a shore.

We created a spatial distribution dataset of the power plant generation capacity allocated to a 5′ × 5′ grid by

combining World Electric Power Plants Database (WEPP) and Carbon Monitoring for Action (CARMA) data to
calculate the electricity generation capacity ratio in each AIM/CGE region. The WEPP (UDI, 2014) provides power
plant name, installed electricity generation capacity, energy source, cooling system type and other information of
power plants around the world. The WEPP includes over 90,000 power plants; however, it does not cover geographic
location of power plants. To determine the geographic coordinates of the power plants, we used the CARMA database

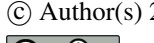



(Center for Global Development, 2014), which contains information on power plant names, carbon emissions, and
geographic coordinates and includes over 60,000 power plants.
Power plants that use seawater for electricity generation must be located adjacent to saline water bodies. Initially,
we assumed that seawater was used for electricity generation if the power plant was located within one grid cell from
a shore. However, the resulting seawater-based generation capacity ratios were too small. For example, almost all
power plants in Japan use seawater; however, under our assumption, only 44% of the generation capacity of coal
power plants was assumed to use seawater. Therefore, we altered this assumption to include power plants located
within two grid cells from a shore. Table 1 shows the electricity generation capacity ratio of seawater-based power
plants to the total electricity generation capacity by AIM/CGE region. The generation capacity ratio was assumed to
be constant over the target period.
The generation capacity ratio has uncertainty, because we only identified locations for half of the power plants
listed in the WEPP. In addition, in developing countries, there are few power plants. For instance, there is only one
nuclear power plant on the African continent, Koeberg Nuclear Power Station. Because this plant was recognized as
a seawater-based power plant under our assumption, all nuclear power plants in Africa were assumed to be seawater-
based over time.

**2.2.    Water use intensity**

We used the water use intensity of each energy source (Table 2) according to Kyle et al. (2013), which essentially
followed that of Macknick et al. (2011). Macknick et al. (2011) presented the minimum, median, and maximum water
use intensity, while Kyle et al. (2013) used median water use intensity derived from Macknick et al. (2011), with
adjustments to previous estimations of electricity sector water use.
The water use intensity of CCS has a high uncertainty, because CCS is a new technology that is not widespread.
Kyle et al. (2013) determined that the water use intensities of coal, integrated coal gasification combined cycle, and
natural gas combined cycle power plants with CCS were about 20–100% higher than those without CCS. However,
they did not include the water use intensities of oil, natural gas, and biomass power plants with CCS. Therefore, we
assumed that the intensities of oil, natural gas, and biomass power plants were 30% higher than those without CCS.
For example, the water withdrawal intensity of oil and natural gas with CCS would be 198 $m^3 MWh^{-1}$, or 30% higher
than that without CCS, 152 $m^3 MWh^{-1}$. We calculated water use by assuming that the water use intensities of plants
with CCS were 100% higher than those without CCS. The impacts of the water use intensities of CCS are discussed
further in Sect. 4.4.
The water consumption intensity of hydropower is controversial. It is difficult to estimate the proportion of water
that evaporates from dams due to hydropower electricity generation, so the water consumption intensity of
hydropower includes the total evaporation from dams. Therefore, we discussed water consumption excluding
hydropower; however, we compared water consumption with and without hydropower in Sect. 4.5.
In thermal power plants, water is primarily used for cooling. Power plants with cooling systems have the greatest
impact on water use for a given type of thermal energy source (IEA, 2012), and the proportions of cooling system
types in use are important when estimating water use for electricity generation. Section 2.2.1 presents the



characteristics of the types of power plant cooling systems, while Sect. 2.2.2 describes the assumptions on proportions
of cooling system types in use.
**2.2.1.    Open-loop and closed-loop cooling systems**
We focused on two cooling systems, open-loop cooling systems (i.e., once-through cooling systems) and closed-
loop cooling systems (i.e., evaporative cooling systems), because most power plants around the world use one of
these two systems. Open-loop cooling systems withdraw water, pass it through a stream condenser, and directly
discharge the heated water into water body (IEA, 2012). They require considerably more water for withdrawal, but
have lower overall water consumption compared with closed-loop cooling systems. Meanwhile, closed-loop cooling
systems withdraw water and pass it through a stream condenser in the same manner as open-loop cooling systems.
However, the heated water is cooled in a wet tower or pond, and the water not evaporated is reused. (IEA, 2012). In
these systems, water withdrawal is much lower, while water consumption is higher compared with the open-loop
configuration.
In terms of environmental impact, in open-loop cooling systems, the subsequent downstream water discharge is
released at temperatures higher than the ambient water, which can be detrimental to aquatic ecosystems. Conversely,
closed-loop cooling systems reduce the potential risks and environmental impacts. Concerns over water shortages
and environmental impacts have motivated a shift from open-loop cooling systems towards closed-loop cooling
systems.
Dry cooling systems represent another important cooling system, although dry cooling systems comprise a very
small proportion of cooling systems. They use air flow instead of water for cooling, so the water use intensity is
negligible. Dry cooling systems are especially useful in water-stressed regions. However, the cost is much higher and
power plant efficiency is lower than both open-loop and closed-loop cooling systems. In this study, we did not
consider dry cooling systems on the assumption that they are not widespread and their overall impact is small.
**2.2.2.    Assumptions on the proportions of cooling system types in use**
The proportion of open-loop and closed-loop cooling systems in use in the base year (2005) was calculated by
estimating water withdrawal for electricity generation in 2005 from Davies et al. (2013).
Because shifts in future cooling system type proportions have a high uncertainty, we had to make several assumptions
to estimate this parameter. Fricko et al. (2016) assumed that open-loop cooling systems would shift towards seawater-
based cooling and dry cooling systems. However, many other studies assumed that open-loop cooling systems would
shift towards closed-loop cooling systems, reflecting recent trends (see Sect. 2.2.1) (Davies et al., 2013; Kyle et al.,
2013; Hejazi et al., 2014; Bijl et al. 2016).
To address this assumption, we created two cases, the 'recent-trend cooling case' and 'status-quo cooling case',
since we were only interested in examining the likely range of cooling system shift impacts. In the recent-trend
cooling case, we applied an assumption reflecting recent trends, in particular, that open-loop cooling system usage
decreases by 0.4% per year until the share of open-loop cooling system usage decreases to 10%, while closed-loop





cooling system usage increases by 0.4% per year until the share of closed-loop cooling system usage increases to
90%. Meanwhile, in the status-quo cooling system case, we assumed that the cooling system type share was fixed to
that of the base year (2005) for comparison with the recent-trend cooling case. In both cases, the proportions of each
cooling system type were the same, regardless of CCS use.
Table 3 lists the proportions of both cooling system types in the recent-trend cooling case for thermal energy
sources from 2005 to 2100, where the proportions of open-loop cooling systems that decreased to 10% and closed-
loop cooling systems that increased to 90% are shaded. Although the change of 0.4% per year was defined arbitrarily,
we used it to represent shifts completed for all thermal energy sources by 2080. Previous studies have also assumed
that cooling system shifts would be completed by the late 21st century (Davies et al., 2013; Fricko et al., 2016). As
the proportion of open-loop cooling systems is unlikely to decrease to 0%, we arbitrarily set the lower limit to 10%,
with a corresponding upper limit for closed-loop cooling systems of 90%.

**3. Results**

**3.1.    Comparison of water use for electricity generation under the two cooling system type cases**

Figure 3 shows the global water withdrawal and consumption for electricity generation under the recent-trend
cooling case and status-quo cooling case for the SSPs and climate mitigation scenarios. Figure 3 includes all of the
cooling system type cases and all of the electricity generation scenarios. This section focuses on the impact of cooling
system type on water use for electricity generation.
Water withdrawal and consumption within a given cooling system case had similar values until 2030, regardless
of the SSPs and climate mitigation scenarios. Although they followed different trends after 2030, water withdrawal
and consumption increased from 2005 to 2100 under all cooling system type cases and electricity generation scenarios,
except water withdrawal under SSP1 in the recent-trend cooling case.
In the recent-trend cooling case, water withdrawal in 2100 under SSP1 was 384–514 $km^3$ $yr^{-1}$, which was 0.7–
0.9 times that in 2005 (555 $km^3$ $yr^{-1}$). Water withdrawal in 2100 under SSP2–5 was 785–1070, 580–906, 856–919,
and 1563–2008 $km^3$ $yr^{-1}$, equivalent to 1.4–1.9, 1–1.6, 1.5–1.7, and 2.8–3.6 times that in 2005, respectively. In the
status-quo cooling case, water withdrawal in 2100 under SSP1–5 was 846–1125, 1713–2658, 1005–2226, 2137–
2335, and 3120–5023 $km^3$ $yr^{-1}$, equivalent to 1.5–2, 3.1–4.8, 1.8–4, 3.8–4.2, and 5.6–9 times that in 2005, respectively.
The increase in water withdrawal was suppressed in the recent-trend cooling case compared with the status-quo
cooling case, and water withdrawal in 2100 in the recent-trend cooling case was 0.4–0.6 times that of the status-quo
cooling case.
In the recent-trend cooling case, water consumption in 2100 under SSP1–5 was 48–67, 94–137, 68–117, 100–
107, and 185–255 $km^3$ $yr^{-1}$, equivalent to 1.9–2.6, 3.7–5.4, 2.6–4.6, 3.9–4.2, and 7.3–10 times that in 2005 (26 $km^3$
$yr^{-1}$), respectively. In the status-quo cooling case, water consumption in 2100 under SSP1–5 was 44–61, 85–121, 64–
103, 88–94, and 170–224 $km^3$ $yr^{-1}$, equivalent to 1.7–2.4, 3.3–4.7, 2.5–4.0, 3.4–3.7, and 6.7–8.8 times that in 2005,
respectively. In contrast to water withdrawal, water consumption differed little between the recent-trend cooling case
and status-quo cooling case, and water consumption in 2100 in the recent-trend cooling case was only 1.1 times





higher than that in the status-quo cooling case.

**3.2.  Comparison of water use for electricity generation under different climate mitigation scenarios and**
**different socioeconomic scenarios**

We only examined the impacts of climate mitigation and socioeconomic changes on water withdrawal and
consumption under the recent-trend cooling case, because it was not dependent on the cooling system type case.
Water withdrawal and consumption did not differ much among the climate mitigation scenarios within a given SSP
scenario (Fig. 3). Comparing water withdrawal among the climate mitigation scenarios, the maximum water
withdrawal in 2100 under SSP1–5 was only 1.3, 1.4, 1.6, 1.1, and 1.3 times higher than the minimum water
withdrawal, respectively. The maximum water consumption in 2100 among the climate mitigation scenarios under
SSP1–5 was only 1.4, 1.5, 1.7, 1.1, and 1.4 times higher than the minimum water consumption, respectively.
In contrast, water withdrawal and consumption differed significantly among the SSPs for a given climate
mitigation scenario (Fig. 3). Comparing water withdrawal among the SSPs, the maximum water withdrawal in 2100
under the baseline, 6.0W, 4.5W, 3.4W, and 2.6W cases was 3.9, 2.8, 4.2, 3.8, and 4.1 times higher than the minimum
water withdrawal, respectively. The maximum water consumption in 2100 among the SSPs under the baseline, 6.0W,
4.5W, 3.4W, and 2.6W cases was 3.8, 2.8, 3.9, 3.6, and 3.8 times higher than the minimum water consumption,
respectively.
To compare the water withdrawal and consumption of each SSP, we calculated the average water withdrawal
and consumption of the climate mitigation scenarios under each SSP in 2100. The average water withdrawal in 2100
for the climate mitigation scenarios was about 444, 868, 687, 875, and 1774 $km^3$ $yr^{-1}$ for SSP1–5, respectively. The
average water consumption in 2100 for the climate mitigation scenarios was about 57, 106, 84, 103, and 213 $km^3$ $yr^{-1}$
for SSP1–5, respectively. SSP5 had the largest average water withdrawal and consumption, which was twice that
of the second largest value. The average water withdrawal and consumption of SSP2 and SSP4 were similar and
represented the second largest values. SSP3 has the fourth largest average water withdrawal and consumption. Finally,
SSP1 has the smallest average water withdrawal and consumption.

**4. Discussion**

**4.1.  Impact of cooling system type**

We compared the recent-trend cooling case with the status-quo cooling case to assess the impact of cooling
system shifts discussed in Sect. 3.1. Water withdrawal was much lower in the recent-trend cooling case, while water
consumption was slightly larger than that in the status-quo cooling case (Fig. 3).
The difference between water withdrawal and consumption in the recent-trend cooling case can be explained by
the water use intensity (Table 2). The water withdrawal intensity of the closed-loop cooling system was much smaller
than that of the open-loop cooling system. Therefore, water withdrawal in the recent-trend cooling case, which
represented the shift from open-loop to closed-loop cooling systems, was much smaller than that in the status-quo



cooling case. Conversely, the difference in water consumption intensity between the open-loop and closed-loop
cooling systems was much smaller compared with that of water withdrawal intensity, although the water consumption
intensity of the closed-loop cooling system was larger than that of the open-loop cooling system. Therefore, water
consumption in the recent-trend cooling case was slightly larger than that in the status-quo cooling case.
Recent shifts in the type of cooling system in use suppressed water withdrawal increases compared the status-
quo case. In contrast, water consumption increased overall, regardless of cooling system type. Previous studies have
also predicted an overall increase in water consumption (Davies et al., 2013; Kyle et al., 2013; Hejazi et al., 2014).

### 4.2. Impact of the climate mitigation and socioeconomic scenarios


We compared the water withdrawal and consumption of each SSP and climate mitigation scenario to assess the
impact of climate mitigation changes and socioeconomic changes described in Sect. 3.2. Water withdrawal and
consumption did not differ substantially among climate mitigation scenarios within a given SSP scenario. In contrast,
water withdrawal and consumption differed significantly among SSPs within a given climate mitigation scenario (Fig.

3).

This can be explained by the composition of energy sources. Figure 4 shows global water withdrawal and
consumption under the recent-trend cooling case by energy source in 2100 for the SSPs and climate mitigation
scenarios. Water withdrawal and consumption consisted mostly of coal, natural gas, nuclear, and biomass power,
because these energy sources had considerable demands on water withdrawal and consumption intensity. Similarly,
oil power had a considerable demand on water withdrawal and consumption intensity; however, it did not have a
large effect on water withdrawal and consumption due to the minimal electricity generated by oil power in all
scenarios. For the same reason, geothermal power did not have a large effect on water consumption, although it
showed considerable water consumption intensity. The water withdrawal and consumption intensities of other energy
sources (i.e., solar and wind power) were negligible. Therefore, water withdrawal and consumption relied heavily on
electricity generation from coal, natural gas, nuclear, and biomass power.
Within a given SSP scenario, the electricity generation from these energy sources did not differ substantially
when the difference between power plants with or without CCS was not taken into account in the climate mitigation
scenarios, except the baseline case (Fig. 2). Under stronger mitigation scenarios, water withdrawal and consumption
increased with electricity generation from power plants with CCS. At the same time, water withdrawal and
consumption decreased with increases in electricity generation from renewable energy. The increased water demand
due to the increase in CCS was negated by the decreased water demand due to the increase in renewable energy.
Therefore, water withdrawal and consumption did not differ greatly among the climate mitigation scenarios (Fig. 4).
Comparing the baseline case with other climate mitigation scenarios, the composition of energy sources was almost
the same in SSP1 and SSP4. However, it differed significantly from the other scenarios, as more fossil fuels were
used in SSP2, SSP3, and SSP5. Therefore, SSP1 and SSP4 had nearly the same water withdrawal and consumption
under all climate mitigation scenarios. However, in SSP2, SSP3, and SSP5, only water withdrawal and consumption
in the baseline case was larger than those under the other climate mitigation scenarios.
Within a given climate mitigation scenario, electricity generation from coal, natural gas, nuclear, and biomass





power plants varied widely among the SSPs (Fig. 2), and water withdrawal and consumption differed significantly
among the SSPs (Fig. 4). The composition of energy sources was influenced greatly by socioeconomic changes. Each
SSP is characterized by multiple assumptions related to energy, including energy cost, energy preference, and social
acceptance (Fujimori et al., 2016b). Even though the climate mitigation targets differed, the assumptions of each SSP
did not change. Therefore, the energy sources applied to each SSP essentially did not change, and only low-carbon
energy, such as renewable energy and CCS, changed according to the climate mitigation target. Thus, climate
mitigation changes had little impact on water withdrawal and consumption for electricity generation. This was
because the electricity generation from energy sources requiring a considerable amount of water was similar among
the climate mitigation scenarios compared with the SSPs, and the water increase driven by CCS was compensated
for by the water decrease driven by renewable energy. In contrast, socioeconomic changes had a large impact on
water withdrawal and consumption for electricity generation because the electricity generation of the energy sources
differed widely among the SSPs. The applicability of these results on a regional scale is discussed in Sect. 4.3.

### 4.3.    Impact of the climate mitigation and socioeconomic scenarios by region

Figure 5 shows the regional water withdrawal differences under the recent-trend cooling case in 2100 among
the SSPs and climate mitigation scenarios. In all regions, the differences among the SSPs under a given climate
mitigation scenario were much larger than those among the climate mitigation scenarios under a given SSP scenario.
The regional water consumption in the recent-trend cooling case exhibited the same trend (Fig. S2). This indicated
that the impact of socioeconomic changes was larger than the impact of climate mitigation changes on a regional
scale, as was the case on a global scale. This trend was observed even though the composition of energy sources
differs drastically between the regional and global scales. As an example, we examined the impacts of climate
mitigation and socioeconomic changes in the Middle East.
The composition of energy sources in the Middle East differed completely from that on a global scale. Under
the baseline case, oil and natural gas accounted for about 90% of electricity generation over the target period in SSP2,
SSP3, and SSP5 (Fig. S3). In contrast, renewable energy grew substantially after 2030, accounting for about 50% of
electricity generation in SSP1 and SSP4. In the other climate mitigation scenarios, renewable energy also grew, and
became the major energy source in SSP2, SSP3, and SSP5 (Fig. S4).
Figure 6shows the water withdrawal and consumption under the SSPs and climate mitigation scenarios for the
recent-trend cooling case. Among the climate mitigation scenarios, the maximum water withdrawal in 2100 was 1.3–
2.2 times higher than the minimum water withdrawal, while the maximum water consumption in 2100 was 1.3–2.2
times higher than the minimum water consumption. In contrast, among the SSPs, the maximum water withdrawal in
2100 was 2.7–4.2 times higher than the minimum water withdrawal, and the maximum water consumption in 2100
was 2.7–4.3 times higher than minimum water consumption. Although the composition of energy sources differed
from that on a global scale, the water increase driven by CCS was negated by the water decrease driven by renewable
energy within a given SSP scenario, and the composition of energy sources varied widely among the SSPs within a
given climate mitigation scenario, as was the case on a global scale. Therefore, the impact of socioeconomic changes
was larger than that of climate mitigation changes in the Middle East.





### 4.4. Comparison of water use for electricity generation under different CCS water use intensities

As described in Sect. 2.2, we assumed that the water use intensities of CCS were 30% higher than those without CCS. However, this assumption had a high uncertainty. We compared the water withdrawal and consumption calculated from different water use intensities of CCS (Fig. 7), where CCS-Low represents water withdrawal and consumption calculated using the 30% assumption, while CCS-High represents water withdrawal and consumption calculated using the assumption that the water use intensities of power plants with CSS were 100% higher those without CCS. The water withdrawal and consumption of CCS-Low and CCS-High in SSP1-4.5W, SSP2-6.0W, SSP4-4.5W and all baseline cases were the same or nearly the same, because CCS was not introduced or only minimally introduced. The water withdrawal and consumption of CCS-High in the other scenarios was 1.2–1.4 times larger than that of CCS-Low. In the stronger mitigation scenarios, water withdrawal and consumption were generally larger because CCS was more widespread.

Comparing CCS-High water use among the climate mitigation scenarios, the maximum water withdrawal in 2100 was 1.1–1.4 times higher than the minimum water withdrawal, while the maximum water consumption in 2100 was 1.1–1.4 times higher than minimum water consumption. In contrast, comparing CCS-High water use among the SSPs, the maximum water withdrawal in 2100 was 2.8–5.5 times higher than minimum water withdrawal, and the maximum water consumption in 2100 among the SSPs was 2.8–5.2 times higher than minimum water consumption. Therefore, the water withdrawal and consumption of CCS-High did not differ greatly among the climate mitigation scenarios, but differed significantly among the SSPs. If the water use intensity of power plants with CCS was 100% higher than those without CCS, the impact of socioeconomic changes would be larger than that of climate mitigation changes.

### 4.5. Comparison of water consumption for electricity generation with/without hydropower

As described in Sect. 2.2, we excluded water consumption from hydropower in this study. To support this decision, we compared water consumption with and without hydropower (Fig. 8). Water consumption with hydropower was more than two times greater than that without hydropower under all scenarios, except SSP5, although the impact of hydropower on water consumption differed among the SSPs. For example, the water consumption of SSP1 and SSP3 was about three times larger with hydropower, because these scenarios had larger electricity generation shares from hydropower. In contrast, the water consumption of SSP5 was only about 1.5 times larger with hydropower, because SSP5 had a small electricity generation share from hydropower. The impact of socioeconomic changes on water consumption for electricity generation was larger than that of climate mitigation changes, regardless of whether hydropower was included.

### 5. Conclusions

This study projected the global water use for electricity generation from 2005 to 2100 for 17 global sub-regions



using the latest scenarios on global change, SSPs which determine socioeconomic conditions and climate mitigation
scenarios which determine climate policies such as targets of GHG emissions. We assessed the impact of shifts in
the proportions of cooling system types in use, as well as the impacts of socioeconomic and climate mitigation
changes.
The results showed that a shift in cooling system types in use resulted in the suppression of water withdrawal
increases in the future compared with the status-quo case. However, water consumption increased regardless of a
shift in cooling system type.
Second, we found that water use differed significantly among the SSPs, because the electricity generation from
energy sources requiring a considerable amount of water varied widely among the SSPs. In contrast, water use did
not differ substantially among the climate mitigation scenarios although they are determinants of GHG emissions,
because the electricity generation from the various energy sources differed less among the climate mitigation
scenarios compared with the SSPs. At the same time, water use increases driven by an increase in the proportion of
power plants with CCS were negated by water use decreases driven by the increased use of renewable energy.
Therefore, socioeconomic changes were predicted to have a much larger impact on water use for electricity generation
compared with climate mitigation changes. Even though the composition of energy sources differed among regions,
this trend was applicable on a regional scale.
We focused on the impact of energy generation on water use. However, water condition (e.g., water scarcity and
increases in water temperature) can also impact electricity generation. For example, water scarcity could constrain
electricity generation from energy sources that require large amounts of water, and increases in water temperature
could reduce power plant efficiency (van Vliet et al., 2016). Such feedback between energy and water should be
taken into account in future predictions. Moreover, tradeoffs between other water users, such as agriculture,
manufacturing, and domestic users, should be considered. To address these challenges, additional studies based on
global hydrological models are necessary to compliment the results of this study.

**Acknowledgments**
This study was supported by JSPS KAKENHI Grant Numbers JP16H06291, JP15H04047; SOUSEI Program
from the Ministry of Education, Culture, Sports, Science and Technology (MEXT); and the Environment Research
and Technology Development Fund (S-10) of the Ministry of the Environment, Japan.

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





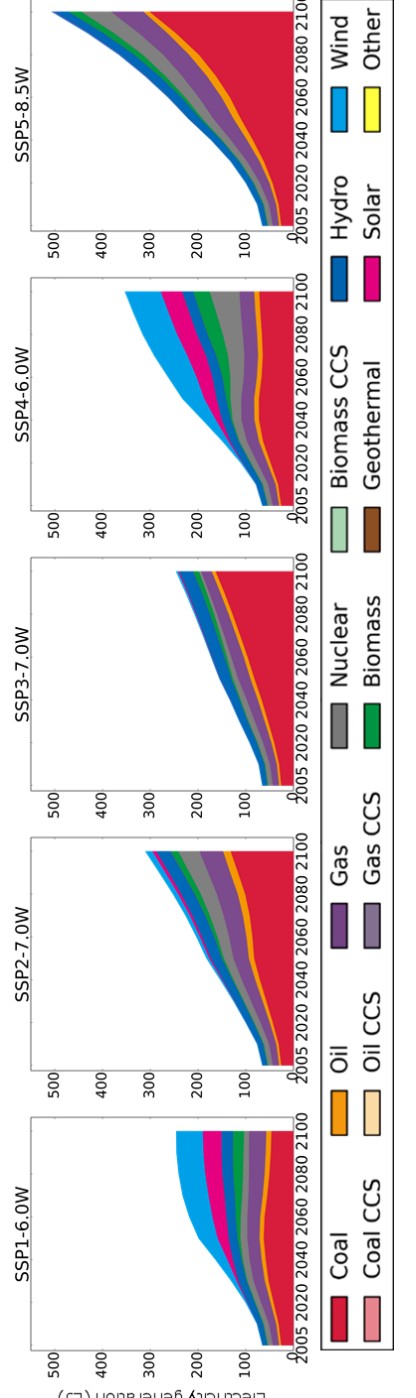

Figure 1 Global electricity generation (EJ) by energy source under the baseline case for the Shared Socioeconomic Pathways (SSPs).





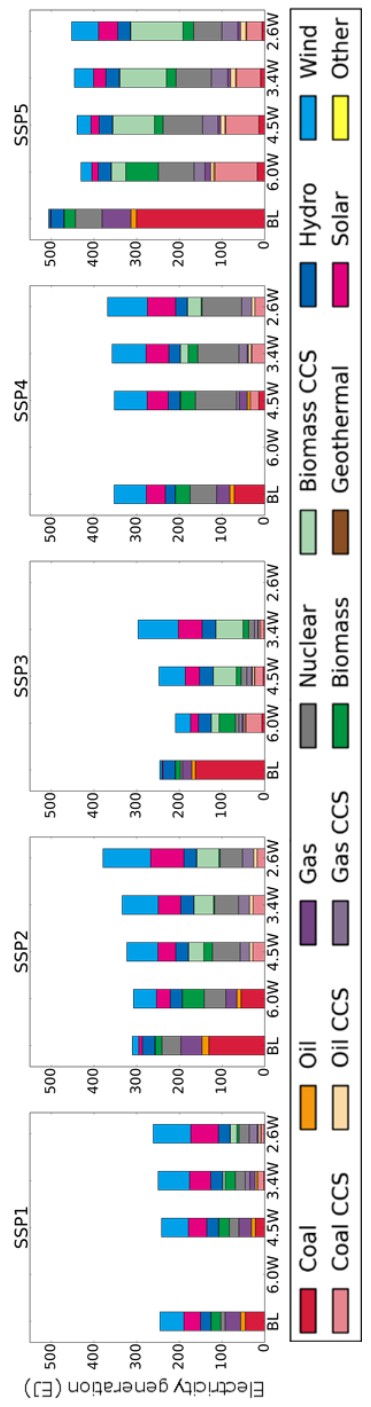

Figure 2 Global electricity generation (EJ) by energy source in 2100 for the SSPs and climate mitigation scenarios. BL, baseline case.





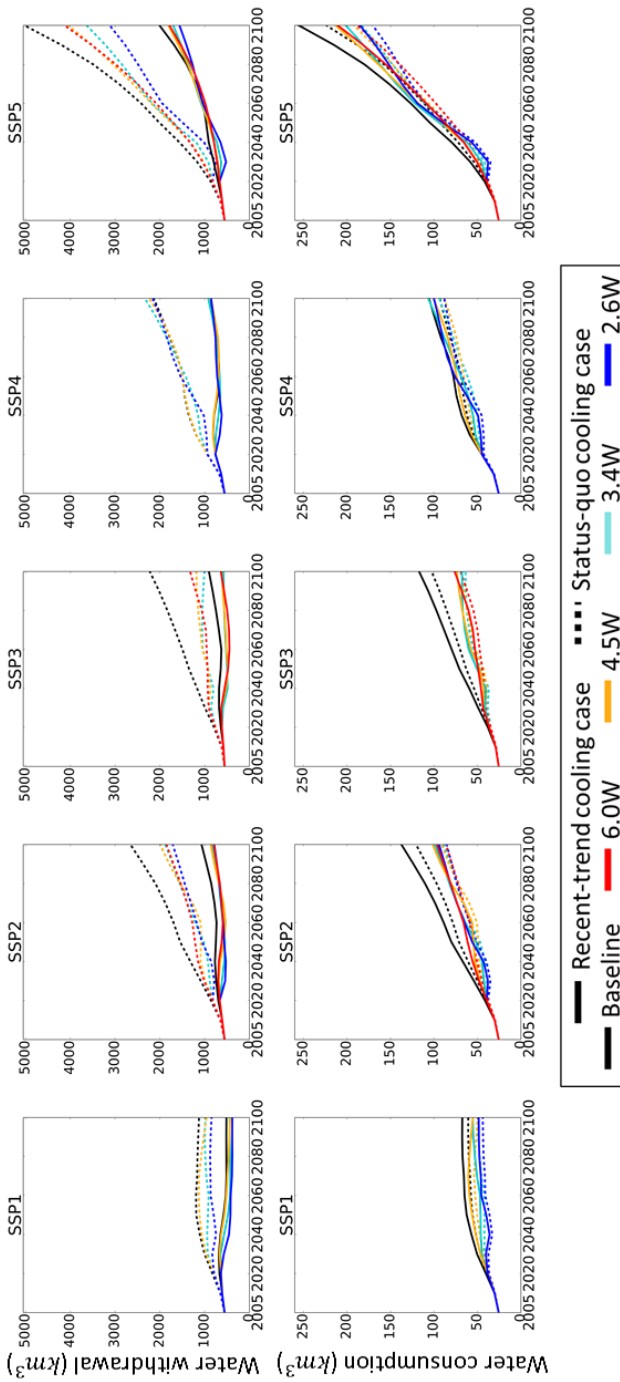

Figure 3 Global (a) water withdrawal (km³ yr⁻¹) and (b) consumption (km³ yr⁻¹) under the recent-trend cooling case and status-quo cooling case for the SSPs and climate mitigation scenarios. The baseline case represents a climate mitigation scenario with no constraints on greenhouse gas (GHG) emissions.




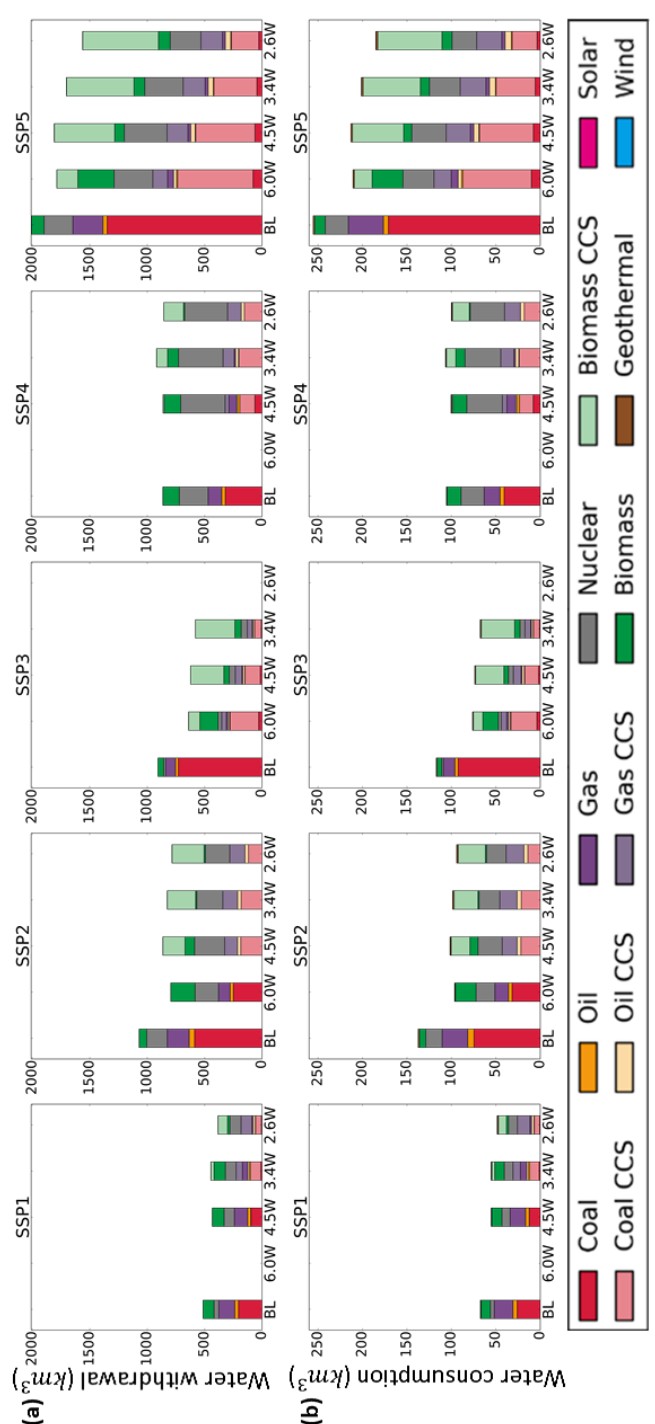

Figure 4 Global (a) water withdrawal (km³ yr⁻¹) and (b) consumption (km³ yr⁻¹) by energy source under the recent-trend cooling case in 2100 for the SSPs and climate mitigation scenarios. BL, baseline case.





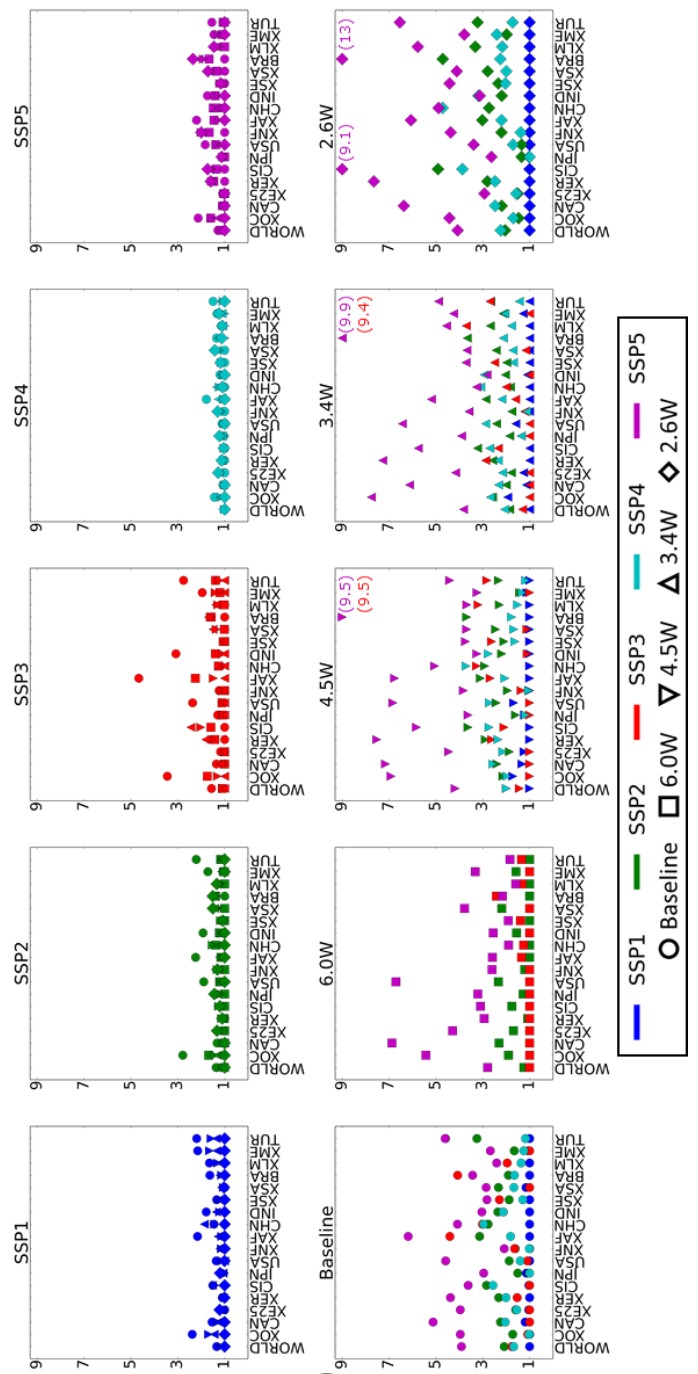

Figure 5 Regional water withdrawal differences under the recent-trend cooling case when (a) the SSP is fixed and (b) the climate mitigation scenario is fixed in 2100. These values were calculated from the water withdrawal of each scenario divided by the minimum water withdrawal among the scenarios. "1" represents the minimum water consumption among the climate mitigation scenarios in (a) and among the SSPs in (b). Values > 9 are plotted at 9 and noted in parentheses.





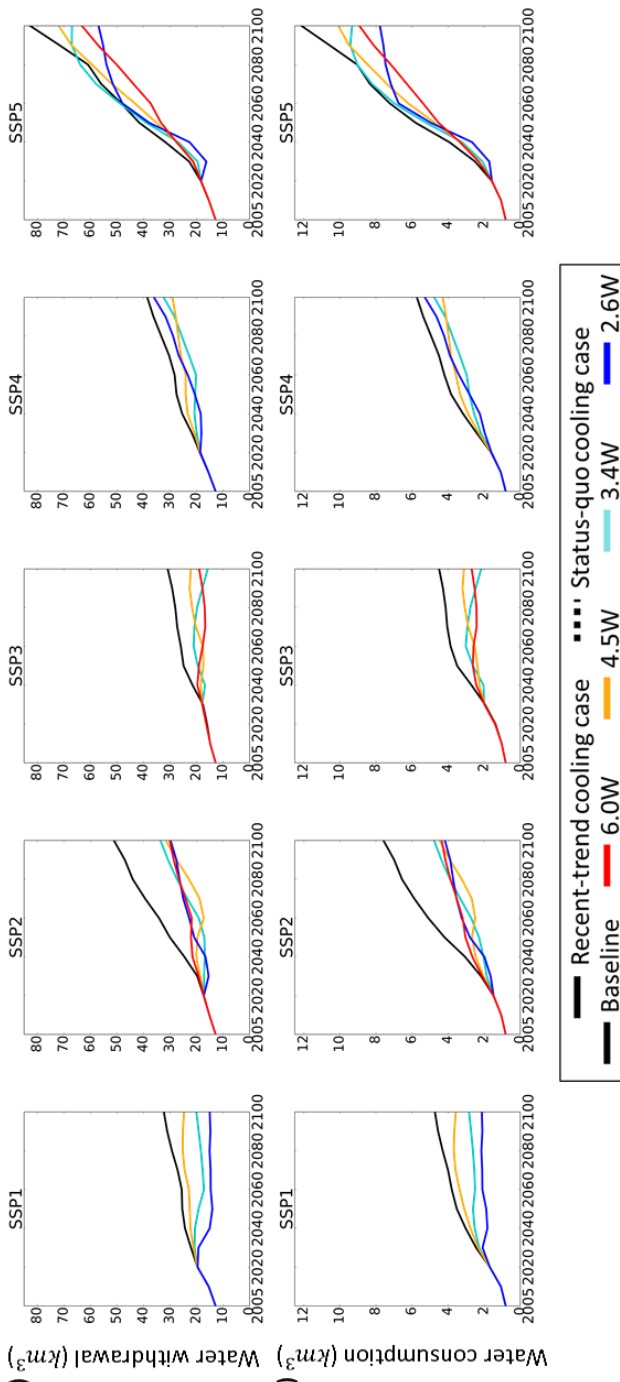

Figure 6 (a) Water withdrawal (km³ yr⁻¹) and (b) consumption (km³ yr⁻¹) under the recent-trend cooling case for the SSPs and climate mitigation scenarios in the Middle East.





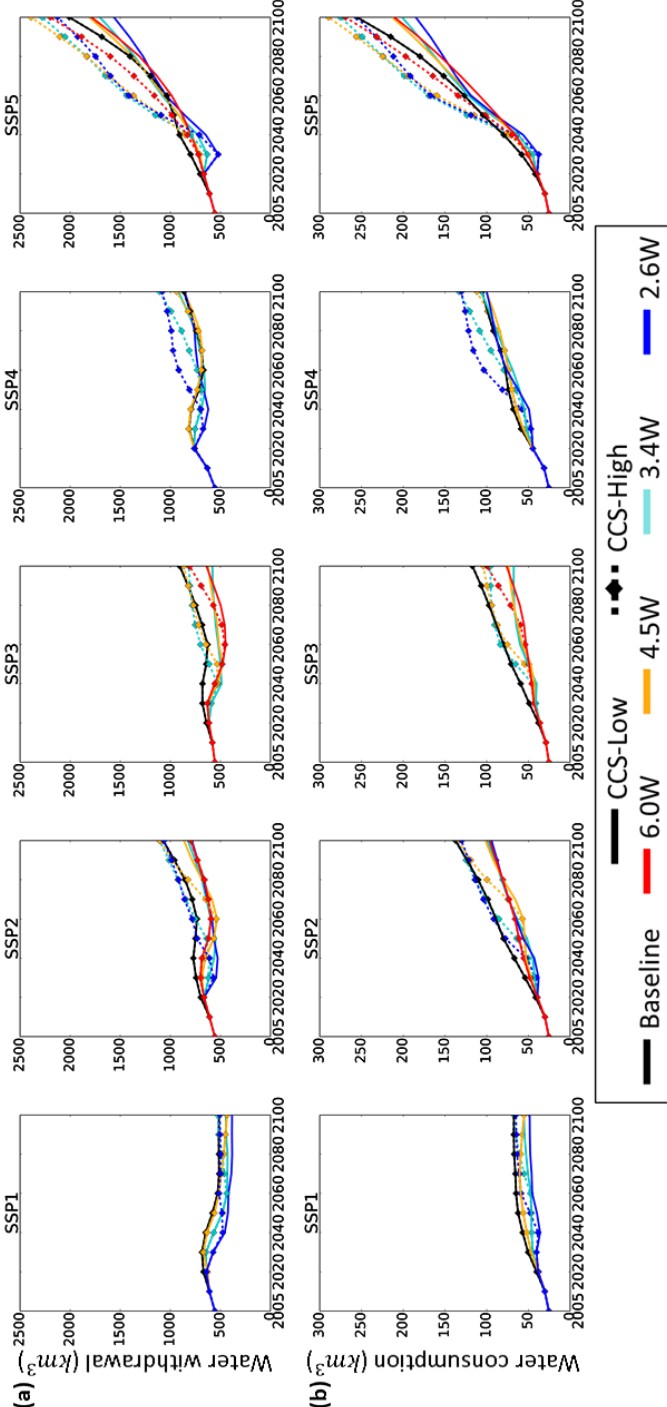

Figure 7 (a) Water withdrawal (km³ yr⁻¹) and (b) water consumption (km³ yr⁻¹) under the recent-trend cooling case for the SSPs and climate mitigation scenarios. CCS-Low and CCS-High represent cases in which water withdrawal and consumption were calculated using assumptions that power plants with CCS had 30% and 100% higher water use intensities, respectively, than without CCS.





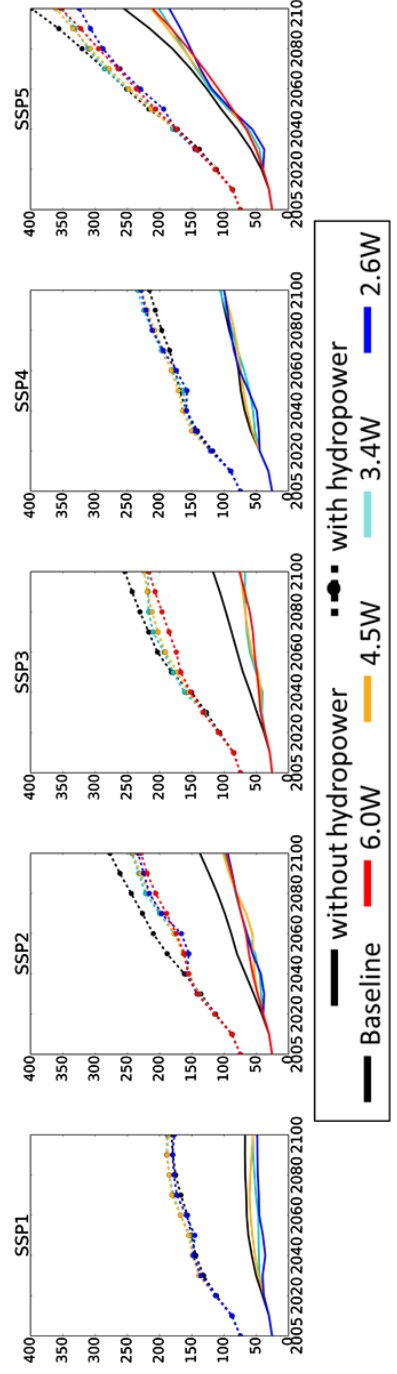

Figure 8 Global water consumption with and without hydropower (km$^3$ yr$^{-1}$) under the recent-trend cooling case (km$^3$ yr$^{-1}$) for the SSPs and climate mitigation scenarios.





Table 1 Electricity generation ratio of seawater-based power plants to total electricity generation (%) by AIM/CGE region.

| No. | Region | Region code | Coal | Oil | Natural gas | Nuclear | Biomass |
|---|---|---|---|---|---|---|---|
| 1 | Oceania | XOC | 14 | 37 | 34 | 0 | 48 |
| 2 | Canada | CAN | 12 | 54 | 5 | 8 | 10 |
| 3 | EU25 | XE25 | 16 | 46 | 29 | 34 | 22 |
| 4 | Rest of Europe | XER | 6 | 16 | 8 | 0 | 1 |
| 5 | Former Soviet Union | CIS | 2 | 1 | 6 | 0 | 1 |
| 6 | Japan | JPN | 75 | 71 | 58 | 100 | 32 |
| 7 | United States | USA | 2 | 22 | 16 | 15 | 11 |
| 8 | North Africa | XNF | 92 | 76 | 51 | 0 | 0 |
| 9 | Rest of Africa | XAF | 3 | 45 | 36 | 100 | 33 |
| 10 | China | CHN | 12 | 15 | 16 | 30 | 8 |
| 11 | India | IND | 12 | 33 | 15 | 29 | 5 |
| 12 | Southeast Asia | XSE | 65 | 58 | 40 | 75 | 27 |
| 13 | Rest of Asia | XSA | 43 | 19 | 21 | 5 | 23 |
| 14 | Brazil | BRA | 17 | 34 | 27 | 100 | 7 |
| 15 | Rest of South America | XLM | 42 | 55 | 37 | 63 | 30 |
| 16 | Middle East | XME | 63 | 47 | 41 | 34 | 42 |
| 17 | Turkey | TUR | 27 | 50 | 53 | 100 | 69 |





Table 2 Water use intensity ($m^3$ $MWh^{-1}$) by energy source and cooling system type, with (w) and without (w/o) carbon capture and storage (CCS).

| Energy source | Cooling system | CCS | Water withdrawal ($m^3$ $MWh^{-1}$) | Water consumption ($m^3$ $MWh^{-1}$) |
|---|---|---|---|---|
| Coal | Open-loop | w/o | 158 | 0.95 |
| | | w | 241 | 1.25 |
| | Closed-loop | w/o | 3.8 | 2.60 |
| | | w | 4.83 | 3.57 |
| Oil/Natural gas | Open-loop | w/o | 152 | 0.91 |
| | | w | 198 | 1.18 |
| | Closed-loop | w/o | 4.55 | 3.13 |
| | | w | 5.92 | 4.07 |
| Nuclear | Open-loop | | 193 | 1.02 |
| | Closed-loop | | 4.17 | 2.54 |
| Biomass | Open-loop | w/o | 152 | 1.14 |
| | | w | 198 | 1.48 |
| | Closed-loop | w/o | 3.32 | 2.09 |
| | | w | 4.32 | 2.72 |
| Geothermal | Closed-loop | | 6.82 | 6.82 |
| Hydro | | | 0 | 17 |
| Solar | | | 0 | 0 |
| Wind | | | 0.02 | 0.02 |


Table 3 Proportion of cooling system type in use (%) by thermal energy source under the recent-trend cooling case from 2005 to 2100.

| Energy source | Cooling system | 2005 | 2010 | 2020 | 2030 | 2040 | 2050 | 2060 | 2070 | 2080 | 2090 | 2100 |
|---|---|---|---|---|---|---|---|---|---|---|---|---|
| Coal | Open-loop | 30 | 28 | 24 | 20 | 16 | 12 | 10 | 10 | 10 | 10 | 10 |
| | Closed-loop | 70 | 72 | 76 | 80 | 84 | 88 | 90 | 90 | 90 | 90 | 90 |
| Oil | Open-loop | 31 | 29 | 25 | 21 | 17 | 13 | 10 | 10 | 10 | 10 | 10 |
| | Closed-loop | 69 | 71 | 75 | 79 | 83 | 87 | 90 | 90 | 90 | 90 | 90 |
| Natural gas | Open-loop | 20 | 18 | 14 | 10 | 10 | 10 | 10 | 10 | 10 | 10 | 10 |
| | Closed-loop | 80 | 82 | 86 | 90 | 90 | 90 | 90 | 90 | 90 | 90 | 90 |
| Nuclear | Open-loop | 38 | 36 | 32 | 28 | 24 | 20 | 16 | 12 | 10 | 10 | 10 |
| | Closed-loop | 62 | 64 | 68 | 72 | 76 | 80 | 84 | 88 | 90 | 90 | 90 |
| Biomass | Open-loop | 14 | 12 | 10 | 10 | 10 | 10 | 10 | 10 | 10 | 10 | 10 |
| | Closed-loop | 86 | 88 | 90 | 90 | 90 | 90 | 90 | 90 | 90 | 90 | 90 |