# Peer review of "Long-term projections of global water use for electricity generation under the Shared Socioeconomic Pathways and climate mitigation scenarios"

_Hydrology and Earth System Sciences, 2017_

## Referee Comment (RC1) · Anonymous Referee #1 · 12 Apr 2017

The paper presents a global analysis of future water requirements for electricity generation. From my perspective this a high-quality paper, but the results have been reported in many previous studies and therefore the scientific significance is not completely clear to me.

The major drawback with the presented analysis and other previous global assessments is that feedbacks between future cooling technology choices and power generation costs are excluded. Switching to more water-efficient cooling technologies in the electricity sector is generally associated with an increase in cost and reduction in energy efficiency. These costs should be included in the CGE model calculations used to project the electricity generation mixtures in each AIM model region to understand

how water constraints impact the mitigation pathways.

---

## Referee Comment (RC2) · Anonymous Referee #2 · 17 May 2017

The authors project electric-sector water withdrawal and consumption for the five Shared Socioeconomic Pathways (SSP) in combination with six climate change mitigation scenarios based on Representative Concentration Pathways (RCP). For each combination of SSP and RCP, the authors used two combinations of cooling systems projections: the status quo, which retains current cooling system shares, and recent trends, which continues current trends towards lower once-through cooling system percentages and much higher recirculating cooling percentages. They compare the differences in water withdrawal and consumption values globally to 2100 among the SSPs versus among the RCP-based mitigation scenarios, and find that water use is much more sensitive to differences among SSPs than RCPs. They also examine the Mid-

dle East as a case study of differences among scenarios, as well as uncertainties in Carbon Capture and Storage (CCS)-based water requirements.

Overall, the paper is well-written and interesting, and will likely be of interest to a wide audience. However, with one important exception, the work is unfortunately basically a repetition using the AIM model of work that has been conducted using other integrated assessment models. The exception relates to the establishment of cooling system types based on the GIS analysis with the WEPP and CARMA databases – this result is interesting and useful in itself, and could be described in greater detail.

Specific comments:

x Line 135: The authors refer to an earlier paper, Fujimori et al. (2016b), for the details of the SSP scenarios. It would be helpful to provide more explanation in the present manuscript as well.

x Section 2.2: The authors assign fixed coefficients for water withdrawal and consumption over the simulation period, but do not explain why they omit technological change that could lower these coefficients over time. Further, in section 2.2.1, the authors explain that they omit dry cooling, because such systems are currently not widespread. This argument is problematic, since adoption of dry cooling is rising, and concerns over water scarcity are likely to drive even wider adoption into the future.

x Line 438-440: The two omissions listed here are significant: water scarcity is not included because of a lack of a global hydrological model, and trade-offs among various water sectors are not included. In my view, there is unfortunately limited value – given the number of other recent studies published on this topic – in presenting projections of electric-sector water use without these feedbacks.

---

## Editor Comment (EC1) · P. van der Zaag (Editor) · 2 Jun 2017

Long-term projections of global water use for electricity generation under the Shared Socioeconomic Pathways and climate mitigation scenarios by Nozomi Ando, Sayaka Yoshikawa, Shinichiro Fujimori, and Shinjiro Kanae

An additional comment by the associate editor

In addition to the comments made by the two anonymous reviewers, with which I agree, I would like to add one additional comment, namely on hydropower. I invite the authors to take this additional comment into consideration when revising the manuscript.

[Figure]

Lines 184-187 The first sentence of this paragraph formulates the water consumption of hydropower too simplistically and incorrectly. It assumes that this consumption is to be equated with "the water that evaporates from dams". This is not correct. First, it should refer to the net evaporation (evaporation from the surface area of the reservoir minus the rainfall on it; so this could be, theoretically as well as in some real cases, a negative value!). Second it should refer to the additional net evaporation compared to the situation without the reservoir (background net evaporation). Third, it should also, and herein lies the complexity, estimate the impact that the change in the timing of water releases from the reservoir due to electricity generation has on the water demands of users located downstream of the dam, including the water demands of aquatic ecosystems.

Although in the literature there is an on-going debate (please provide some key references, e.g. Grubert, 2016; Spang et al., 2014; Scherer and Pfister, 2016, etc.), the issue is in fact quite straightforward and in my view not "controversial" (line 184). What is true is that, given the above, it is not a trivial exercise to accurately estimate the water consumption of hydropower, and most likely requires not only a multidisciplinary approach, but also a basin-wide approach. I do not know whether this is a sufficient argument for the authors to (happily?) decide to leave water consumed for hydropower out of the assessment, which is stated in the second and last sentence of this paragraph.

Section 4.5 (lines 406-414) Given the above it remains unclear how Figure 8 has been constructed. How has water consumption with hydropower been estimated? I have my doubts whether indeed "water consumption with hydropower was more than two times greater than without hydropower" (lines 407-408). So my guess is that it will be less, but very likely still a significant proportion of all water consumed.

If it is indeed a significant water consumer, then was it a defendable choice to leave hydropower out of the analysis? I think not. And why leave it out if, in the end you nevertheless present results that include this important water user?
I invite the authors to carefully consider the reviews by the two anonymous reviewers, as well as the above point on hydropower, a point not raised by them.

Pieter van der Zaag
* * *

---

## Author Comment (AC1) · 15 Jun 2017

**Author's response to Reviewers and associated editor's comments**

We would like to submit our response for our manuscript entitled "Long-term projections of global water use for electricity generation under the Shared Socioeconomic Pathways and climate mitigation scenarios [Doi: 10.5194/hess-2017-27]" submitted to Hydrology and Earth System Sciences

First of all, we would like to express here to each one of them, our deepest gratitude for your invaluable critics, comments and suggestions that, we sincerely believe, helped us revise our paper, trim it, and streamline its contents and formulation, so as to respond to high requirements for publication.

With our best regards, and our deepest gratitude to you and to all the reviewers and associated editor.

Sincerely yours
Sayaka Yoshikawa (on behalf of authors)

**Response to Referee #1**

**Comment 1 :**

The paper presents a global analysis of future water requirements for electricity generation. From my perspective this a high-quality paper, but the results have been reported in many previous studies and therefore the scientific significance is not completely clear to me.

> **Response :**
>
> We would like to express here our deep gratitude for the invaluable comments that are helpful for us to revise our manuscript. We are in the process of addressing all of the comments accordingly, revise and submit our manuscript.
>
> Our scientific significance of this study is that this is the first to estimate water use of electricity generation for cooling with incorporating energy-related factors under the SSPs and the RCPs which were used in the IPCC AR5 as latest scenarios. Similar to this study, previous studies on a global scale assessed the impacts of demand drivers that affect water use for electricity generation (Davies et al. 2013, Kyle et al. 2013 and Hejazi et al. 2014).
>
> There were several differences in the methods between this study and these previous studies, including the future scenarios. One of the largest differences was that we took both socioeconomic scenarios represented by the SSPs and climate mitigation scenarios based on RCPs into account in our simulation. As a consequence, we revealed that the socioeconomic changes had a larger impact on water withdrawal and consumption of electricity generation for cooling, compared with the climate mitigation changes. At the beginning of this study, authors had expected that climate mitigation changes were larger impact on water use for cooling than socioeconomic impact because energy consumption decrease

through energy saving for mitigation. These results were able to be accomplished by projection of water use under both climate change and socioeconomic scenarios.

Other similar to this study, there are Fujimori et al. (2017). We used same electricity generation data as Fujimori et al (2017), who is one of co-authors in this study; however, our results were not the same. Fujimori et al. (2017) applied technological improvement rates to water use intensity. They assumed that the technological improvement rates were consistent with SSP narratives. However, they found that future technological improvements were difficult to predict and had large uncertainties. We made assumptions on the shift in the proportion of cooling system types in use to represent one technological improvement.

In addition, there are additional scientific significance of this study. We originally created database of electricity generation ratio (of freshwater and seawater use) of electricity generation plants by using integrated geographic information which were combined geographic location from CARMA and plant information details from WEPP database. There is no other data like this.

The significance has added in revised manuscript.

**Comment 2 :**
The major drawback with the presented analysis and other previous global assessments is that feedbacks between future cooling technology choices and power generation costs are excluded. Switching to more water-efficient cooling technologies in the electricity sector is generally associated with an increase in cost and reduction in energy efficiency. These costs should be included in the CGE model calculations used to project the electricity generation mixtures in each AIM model region to understand how water constraints impact the mitigation pathways.

> **Response :**
> Thank you very much for your helpful comments. We agree with you that switching to more water-efficient cooling technologies in the electricity sector is generally associated with an increase in cost and reduction in energy efficiency. Actually, Turchi et al. (2010) suggested that the performance penalty for concentrating Solar Power facilities switching from wet cooling to dry cooling results in an annual reduction in output of 2%–5% and an increase in the levelized cost of producing energy of 3%–8%, depending on local climatic conditions.
>
> As you pointed out, these costs included in our CGE model can be technically computable. However, concerns remain about uncertainty in limited data; for example, current energy efficiency and ratios of cooling-cost to total cost of power generation in each power plant, cooling system share in each country. In fact, while the cooling system share for the base year of 2005 in the United States, Australia and China were available in the literature, shares in all other region were

estimated in Davies et al. (2013). If we will continue to use these assumptions in the future simulation, it would entail large uncertainties. Even if the power generation associated with increase in cost and reduction in energy efficiency is estimated in the CGE model, we expect our revelation might not be changed. For establishing more realistic estimation, we will hope to investigate the feedback between future cooling technology choices and power generation costs more detail as our future tasks. These explanations have been added in discussion part of the revised manuscript.

**Response to Referee #2**

**Comment 1**:

Overall, the paper is well-written and interesting, and will likely be of interest to a wide audience. However, with one important exception, the work is unfortunately basically a repetition using the AIM model of work that has been conducted using other integrated assessment models. The exception relates to the establishment of cooling system types based on the GIS analysis with the WEPP and CARMA databases – this result is interesting and useful in itself, and could be described in greater detail.

**Response**:

First of all, we would like to express here our deep gratitude for the invaluable comments that are helpful for us to revise our manuscript. Here, our results were not able to be achieved using only output of the AIM model. We just used electricity generation from each supply source which are output from the AIM model. Then, the impacts of socioeconomic and climate mitigation changes on water use of each electricity generation were calculated using the output, water use intensity by energy source and originally created database of electricity generation ratio which were created by WEPP and CARMA.

Similar to this study, previous studies on a global scale assessed the impacts of demand drivers that affect water use for electricity generation (Davies et al. 2013, Kyle et al. 2013 and Hejazi et al. 2014). These previous studies applied the same methodology over a target period of 2005–2095 using the Global Change Assessment Model, an integrated assessment model. There were several differences in the methods between this study and these previous studies, including the future scenarios. One of the largest differences was that we took our simulation into account under both socioeconomic scenarios represented by the SSPs and climate mitigation scenarios based on RCPs. We revealed that the socioeconomic changes had a larger impact on water withdrawal and consumption of electricity generation for cooling, compared with the climate mitigation changes. Before starting this study, at least, authors had expected that climate mitigation changes were larger impact on water use than socioeconomic impact because energy consumption decrease through energy saving for mitigation. This

results were able to be accomplished by projection of water use under both climate change and socioeconomic scenarios.

We have modified for making difference between this study and previous study more clear in revised manuscript. We have added more detailed explanation of the establishment of cooling system types based on the GIS analysis in Sec 2.1.2 of revised manuscript according to your comment.

**Comment 2**:

x Line 135: The authors refer to an earlier paper, Fujimori et al. (2016b), for the details of the SSP scenarios. It would be helpful to provide more explanation in the present manuscript as well.

> **Response**:
>
> Thank you for your suggestion. More details explanation of Fujimori et al. about scenarios have been added in method and data part (Sec 2.1.1) of revised manuscript.

**Comment 3**:

x Section 2.2: The authors assign fixed coefficients for water withdrawal and consumption over the simulation period, but do not explain why they omit technological change that could lower these coefficients over time. Further, in section 2.2.1, the authors explain that they omit dry cooling, because such systems are currently not widespread. This argument is problematic, since adoption of dry cooling is rising, and concerns over water scarcity are likely to drive even wider adoption into the future.

> **Response**:
>
> In previous study, Fujimori et al. (2017) mentioned that technological progress associated with time was statistically significant in the panel data model analysis of historical data, but it is highly uncertain whether these results can be extrapolated indefinitely into the future. In this study, we made assumptions on the shift in the proportion of cooling system types in use to represent one technological improvement. As we mentioned in S3.1 of supplemental, we did not take other technological improvements into account, such as power plant efficiency improvements or the introduction of water-saving technologies, to avoid increasing the uncertainty. Our water withdrawal and consumption results were generally larger than those of Fujimori et al. (2017).
>
> In terms of dry cooling, our simulation was not taken dry cooling into account throughout overall period. It is of course important and useful technology in water-stressed regions. The dry cooling method uses the natural air present in the atmosphere for cooling instead of water. Water consumption will be minimal in the dry cooling. In theory, dry cooling is relatively feasible as an option for new generating plants of all kinds, in relation to its application as a retrofit on existing

facilities. For taking dry cooling share into account in our simulation, breakthroughs need to be made to the following concerns. First, the released heat to atmosphere would help enhance greenhouse effect. Our climate mitigation had not included in the impact. Second, NETL (2011) mentioned that relevant for addressing any water-related constraints, most thermoelectric power generation technologies can use dry cooling systems, which have low water demands compared with water-flow-based cooling systems, but incur higher capital costs, lower thermal efficiencies, and lower power output, all of which would increase the total costs of electricity generation as same as first referee pointed out. These costs should be included in the CGE model calculations used to project the electricity generation mixtures in each region. Third, the share of dry cooling is still a few. The data is very limited. Actually, a 2006 Department of Energy (DOE) reported in the USA 43% of thermal electric generating capacity uses once-through cooling, 42% wet recirculating cooling, 14% cooling ponds and 1% dry cooling.

We have added this description in the revised manuscript.

**Comment 4 :**

x Line 438-440: The two omissions listed here are significant: water scarcity is not included because of a lack of a global hydrological model, and trade-offs among various water sectors are not included. In my view, there is unfortunately limited value – given the number of other recent studies published on this topic – in presenting projections of electric-sector water use without these feedbacks.

**Response :**

Our objective is to comprehensively estimate water use of electricity generation for cooling under both socioeconomic and climate mitigation scenarios. For understanding each impact, estimation of the impact which excluded other water related impact (e.g. water scarcity and trade-off) is very important. Then, we can take the other water related impact into account based on our study.

In order to taking such water related impact into accounts, calculating by a global hydrological model is mandatory. However, it is challenging task to incorporate water use for electricity generation with the global hydrological model. There are two main reasons. First, the reason is difficulty of combination between CGE model and hydrological model. Research about the combination is still very few. It is ongoing and necessary to work on multiple and great projects. This is completely outside the reach of this study. For example, one of problems still remain unsolved is about spatial resolution. The global hydrological model is a grid scale, but CGE model covered all regions of the world divided into only 17 sub-regions.

Second, the reason is that still there are very few in state-of-the-art hydrological model for taking trade-offs among various water sectors into account. So far, there is only way of projection that how much water we need in the future, namely demand-based estimation. In order to taking the trade-offs into account on a global scale, we need to create new hydrological model that agricultural production and industrial and domestic activity would be restricted if there occur water scarcity. Global hydrological model H08 (e.g. Hanasaki et al. 2017), which were developed by our team, just had been achieved to develop attributing the water sources available to humanity, but it's still not taking trade-off into account. We will hope to consider these water related impacts as next studies including improvement of the hydrological model.

**Response to associate editor Dr. P. van der Zaag**

**Comment 1:**

In addition to the comments made by the two anonymous reviewers, with which I agree, I would like to add one additional comment, namely on hydropower. I invite the authors to take this additional comment into consideration when revising the manuscript.

> **Response:**
> First of all, we would like to express here our deep gratitude for the invaluable comments about hydropower that are helpful for us to revise our manuscript. We are in the process of addressing all of the comments accordingly, revise and submit our manuscript. All of the concerns you raised have now addressed as below.

**Comment 2:**

Lines 184-187 The first sentence of this paragraph formulates the water consumption of hydropower too simplistically and incorrectly. It assumes that this consumption is to be equated with "the water that evaporates from dams". This is not correct. First, it should refer to the net evaporation (evaporation from the surface area of the reservoir minus the rainfall on it; so this could be, theoretically as well as in some real cases, a negative value!). Second it should refer to the additional net evaporation compared to the situation without the reservoir (background net evaporation). Third, it should also, and herein lies the complexity, estimate the impact that the change in the timing of water releases from the reservoir due to electricity generation has on the water demands of users located downstream of the dam, including the water demands of aquatic ecosystems.

> **Response:**
> Thank you so much for your invaluable comments. We agree with your comment that water consumption of hydropower too simplistically and incorrectly. Main target of this study is cooling water of electricity generation in industrial water. on a case-by-case basis, we can remove about the hydropower from the

manuscript. However, we would like to leave behind it as just additional information for readers. We have thoroughly revised manucript (including abstract and title) about our main target according to your comments and added the following description in discussion part of the revised manuscript.

We just followed widely used method from previous studies (DOE 2006, Kyle et al. 2013 and Davies et al. 2013). In the previous studies, water consumption intensity of hydro power assumed that all reservoir evaporation allocates to power production. As you pointed out, it is singularly outstanding problem. This approach certainly does not take into account the evaporation losses prior to construction of hydropower plant. In our estimation, water consumptions of hydropower generation are based on analysis of water evaporation from reservoirs and as such may not accurately bound the range for hydroelectricity-related water evaporation in all regions. Assigning all evaporative losses to electricity generation ignores the fact that dams are often constructed for multiple uses, such as water supply for municipal or agricultural purposes, flood control, and recreation. In addition, change in the timing of water releases from the reservoir and the impact on water user and aquatic ecosystems were not taken into account. If combination between the CGE model and the hydrological model will be successful, these problems would be likely to resolve. More reliable estimates of the water consumption of hydropower would greatly improve any attempts to quantify and attribute water consumption in current and future periods.

**Comment 3 :**
Although in the literature there is an on-going debate (please provide some key references, e.g. Grubert, 2016; Spang et al., 2014; Scherer and Pfister, 2016, etc.), the issue is in fact quite straightforward and in my view not "controversial" (line 184). What is true is that, given the above, it is not a trivial exercise to accurately estimate the water consumption of hydropower, and most likely requires not only a multidisciplinary approach, but also a basin-wide approach. I do not know whether this is a sufficient argument for the authors to (happily?) decide to leave water consumed for hydropower out of the assessment, which is stated in the second and last sentence of this paragraph.

    **Response :**
       Thank you so much for your invaluable comments and providing worthful references. We have revised following directions from your comments.

**Comment 4 :**
Section 4.5 (lines 406-414) Given the above it remains unclear how Figure 8 has been constructed. How has water consumption with hydropower been estimated? I have my doubts whether indeed "water consumption with hydropower was more than two times greater than without hydropower" (lines 407-408). So my guess is that it will be less,

but very likely still a significant proportion of all water consumed. If it is indeed a significant water consumer, then was it a defendable choice to leave hydropower out of the analysis? I think not. And why leave it out if, in the end you nevertheless present results that include this important water user?

**Response:**

We just calculated by multiplying the electricity generation (using Fujimori et al. 2017) and water use intensity of hydropower (using Kyle et al. 2013) as same as calculation of other energy sources. We agree with your comment that water consumption of hydropower generation will be less. It might be changed if outstanding problems which mentioned the above will be resolved.

Hydropower is important users. However, main target of this study is cooling water of electricity generation in industrial water. Generally, industrial water includes water for the cooling of thermoelectric, nuclear power plants etc., but it does not include hydropower (FAO, 2017). We would like to leave behind it as just additional information for readers.

We have thoroughly revised for more easily understandable about our target in revised manuscript including title and abstract.

**References**

FAO.: AQUASTAT database. http://www.fao.org/nr/water/aquastat/data/popups/itemDe fn.htm l?id=4252, 2017.

Davies, E. G., Kyle, P., and Edmonds, J. A.: An integrated assessment of global and regional water demands for electricity generation to 2095. Adv. Water Resour., 52, 296–313, doi:10.1016/ j.advwatres.2012.11.020, 2013.

US Department of Energy. Energy demands on water resources: report to Congress on the interdependency of energy and water. US Department of, Energy; 2006.

Fujimori, S., Hasegawa, T., Masui, T., Takahashi, K., Herran, D. S., Dai, H., Hijioka, Y., and Kainuma, M.: SSP3: AIM implementation of Shared Socioeconomic Pathways. Glob. Environ. Change, doi:10.1016/j.gloenvcha.2016.06.009, 2017.

Hanasaki, N., Yoshikawa, S., Pokhrel, Y., and Kanae, S.: A global hydrological simulation to specify the sources of water used by humans, Hydrol. Earth Syst. Sci. Discuss., https://doi.org/1 0.5194/hess-2017-280, in review, 2017.

Hejazi, M., Edmonds, J., Clarke, L., Kyle, P., Davies, E., Chaturvedi, V., Wise, M., Patel, P., Eom, J., Calvin, K., Moss, R., and Kim, S.: Long-term global water projections using six socioeconomic scenarios in an integrated assessment modeling framework. Technol. Forecast. Soc., 81(1), 205–226, doi:10.1016/j.tec hfore.2013.05.006, 2014.

Kyle, P., Davies, E. G., Dooley, J. J., Smith, S. J., Clarke, L. E., Edmonds, J. A., and Hejazi, M.: Influence of climate change mitigation technology on global demands

of water for electricity generation. Int. J. Greenhouse Gas Control, 13, 112–123, doi:10.1016/j.ijggc.2012.12.006, 2013.

NETL. Reducing Freshwater Consumption at Coal-Fired Power Plants: Approaches Used Outside the United States. DOE/NETL-2011/1493. National Energy Technology Laboratory, Pittsburgh, PA, 2011.

Turchi, C.; Wagner, M.; Kutscher, C. Water Use in Parabolic Trough Power Plants: Summary Results from WorleyParsons' Analyses. NREL/TP-5500-49468. Golden, CO: National Renewable Energy Laboratory, 2010.

---

## Author Comment (AC2) · 15 Jun 2017

Please see attachment.

Please also note the supplement to this comment:
http://www.hydrol-earth-syst-sci-discuss.net/hess-2017-27/hess-2017-27-AC2-supplement.pdf
* * *